# Gene Profile of Adipose Tissue of Patients with Pheochromocytoma/Paraganglioma

**DOI:** 10.3390/biomedicines10030586

**Published:** 2022-03-02

**Authors:** Judita Klímová, Miloš Mráz, Helena Kratochvílová, Zdeňka Lacinová, Květoslav Novák, David Michalský, Jan Kvasnička, Robert Holaj, Denisa Haluzíková, Radka Petráková Doležalová, Matěj Zítek, Zuzana Krátká, Veronika Todorovová, Jiří Widimský, Martin Haluzík, Tomáš Zelinka, Ondřej Petrák

**Affiliations:** 1Center of Hypertension, 3rd Department of Medicine, First Faculty of Medicine and General Faculty Hospital, Charles University, 128 00 Prague, Czech Republic; judita.klimova@vfn.cz (J.K.); jan.kvasnicka3@vfn.cz (J.K.); robert.holaj@vfn.cz (R.H.); matej.zitek@vfn.cz (M.Z.); zuzana.kratka@vfn.cz (Z.K.); jwidi@lf1.cuni.cz (J.W.J.); tzeli@lf1.cuni.cz (T.Z.); 2Center for Experimental Medicine and Diabetes Center, Institute for Clinical and Experimental Medicine, 140 00 Prague, Czech Republic; milos_mraz@yahoo.co.uk (M.M.); lacz@ikem.cz (Z.L.); mhalu@lf1.cuni.cz (M.H.); 3Institute for Medical Biochemistry and Laboratory Diagnostics, First Faculty of Medicine and General Faculty Hospital, Charles University, 128 00 Prague, Czech Republic; krth@ikem.cz; 4Department of Urology, First Faculty of Medicine and General Faculty Hospital, Charles University, 128 00 Prague, Czech Republic; kvetoslav.novak@vfn.cz; 5First Department of Surgery, First Faculty of Medicine and General Faculty Hospital, Charles University, 128 00 Prague, Czech Republic; david.michalsky@vfn.cz; 6Institute of Sport Medicine, First Faculty of Medicine and General Faculty Hospital, Charles University, 128 00 Prague, Czech Republic; denisa.haluzikova@lf1.cuni.cz (D.H.); radka.petrakovadolezalova@vfn.cz (R.P.D.); 7Laboratory of Endocrinology and Metabolism, 3rd Department of Medicine, First Faculty of Medicine and General Faculty Hospital, Charles University, 128 00 Prague, Czech Republic; veronika.todorovova@vfn.cz

**Keywords:** pheochromocytoma, functional paraganglioma, brown adipose tissue, beige adipose tissue, gene expression, metabolism, lipids, energy metabolism

## Abstract

Background: Brown adipose tissue (BAT) is a therapeutic target to combat obesity and related disorders. Pheochromocytoma and functional paraganglioma (PPGL) are associated with activated BAT due to catecholamine excess. Our aim was to evaluate BAT activity by gene profile and assess its relation to clinical characteristics and overproduced catecholamine. Methods: mRNA expression of 15 genes in subcutaneous adipose tissue (SAT) and visceral adipose tissue (VAT) was measured via RT-PCR in 25 patients with PPGL and 14 controls undergoing cholecystectomy. Results: We found in VAT of PPGL higher expression of *UCP1* (*p* < 0.001), *CEBPB*, *PPARGC1A* (both *p* < 0.001), *PRDM16* (*p* = 0.069) and *DIO2* (*p* = 0.005). *UCP1* expression correlated only with norepinephrine levels and its metabolite. *UCP1* expression, among others, correlated negatively with BMI, age and positively with HDLc levels. Dominance of BAT or BeAT markers was not assessed in PPGL. In SAT of PPGL, we found higher expression of *ADRB3*, *CIDEA* (both *p* < 0.05), and *PPARGC1A* (*p* = 0.001), but not *UCP1*. Conclusion: We demonstrate signs of *UCP1*-dependent norepinephrine-induced thermogenesis connected with higher expression of *DIO2*, *PPARGC1A*, *CEBPB* and *PRDM16* in retroperitoneal VAT of PPGL and its relations to circulating HDLc and triglycerides levels. However, no direct relationship with increased basal energy metabolism measured by calorimetry was found.

## 1. Introduction

Pheochromocytomas and functional paragangliomas (PPGL) are rare neuroendocrine tumors arising from chromaffin cells of adrenal medulla or extra-adrenal sympathetic ganglia located in the thorax, abdomen or pelvis [1]. PPGL are characterized by overproduction of one or more catecholamines, which leads to the development of changes in carbohydrate, lipid and whole-body metabolism [2]. Weight loss despite normal appetite and food intake is common in patients with PPGL [3,4], and is caused by a higher metabolic rate in comparison with healthy controls and the state after tumor removal [5,6]. Processes which contribute to the hypermetabolic state include catecholamine-induced glycogenolysis and lipolysis, and overproduction of proinflammatory cytokines [5]. Activation of brown adipose tissue (BAT) and beige adipose tissue (BeAT) as one of the mechanisms of hypermetabolism in PPGL remains controversial. PPGL were recognized as one of the three rare conditions (together with hibernoma and cancer cachexia), with the occurrence of a significant amount of metabolic active BAT in human adults [7,8,9,10]. With the widespread use of nuclear medicine imaging, BAT has been recognized as important metabolic active tissue with possible usage in the treatment of obesity and diabetes mellitus type 2 in the general population.

BAT plays a crucial role in adaptive non-shivering thermogenesis (NST). Glucose and fatty acids (FA) are taken up by the brown and beige adipocytes, and the uncoupling protein 1 (*UCP1*), a marker of BAT, uncouples the oxidative respiration of FA from adenosine triphosphate (ATP) release and leads to the production of heat instead of ATP. Fatty acids are derived from stored triglycerides (TAG) in the process of lipolysis in brown and white adipocytes, and emerge from intestinal chylomicrons. Activated BAT/BeAT attenuates hyperlipidaemia and improves insulin resistance [11]. NST is activated by the sympathoadrenal system, mainly norepinephrine [12]. It was shown in animal models that epinephrine also affects the processes [13]. Those studies showed that after various stimuli, including beta3-adrenoceptor agonists, BeAT appeared in WAT depots, and is also capable of adaptive thermogenesis [14,15]. The origin of beige adipocytes is unclear [16,17]. Many factors were reported in literature as browning agents (e.g., cold exposure, beta3 adrenergic receptor agonists, exercise, hormonal factors, inflammatory factors), but mostly in rodents [16]. In patients with PPGL, the presence of active BAT could be associated with higher mortality [18]. Activation of BAT/BeAT is seen also in cancer cachexia. How cancer triggers BeAT/BAT has not been established yet [19]. BAT/BeAT was also reported as a tissue with endocrine functions [20]. 

Gene expression is one way of discriminating between the distinct types of adipose tissue. The aim of our study was to evaluate an expression profile of genes potentially related to brown and beige adipogenesis, and activation in patients with PPGL. Furthermore, we tried to assess the relationship of these genes, particularly *UCP-1*, to clinical characteristics and overproduced catecholamine. A real-time polymerase chain reaction (PCR) was used for expression of 15 genes of interest and 3 housekeeping genes in samples of whole subcutaneous (SAT) and visceral adipose tissue (VAT) sourced from patients with PPGL and control subjects. Retroperitoneal VAT was chosen as a tissue with generally high content of BAT. Contrarily, omental WAT was chosen as generally BAT-free tissue. Indirect calorimetry was used for quantification of energy metabolism in both groups. Blood samples were taken.

## 2. Materials and Methods

### 2.1. Recruitment and Background

In total, 25 subjects with PPGL (24 subjects with pheochromocytoma and 1 abdominal functional paraganglioma) and 14 controls were included in the study. All patients with PPGL were examined during a short hospitalization in our department (General University Hospital, Prague, Czech Republic). The diagnosis of PPGL was based on free plasma metanephrine levels, visualization of the tumor by computer tomography (CT) or positron emission tomography–computed tomography (PET/CT) with fluorodopa. The diagnosis was confirmed histopathologically. All patients underwent genetic testing for germline mutations. No patient suffered from von Hippel-Lindau syndrome, multiple endocrine neoplasia type 2 or neurofibromatosis type 1. No patient had mutation in the *SDHx* or *MAX* gene. One patient had positive testing for germline mutation in *TMEM127* c.159del (p.Trp53CysfsTer28). All other causes of secondary hypertension were excluded. Control group included patients undergoing elective cholecystectomy in a stable phase of the disease. No signs of inflammation were present. Control subjects were not treated for any severe chronic disease, oncology or otherwise. Their liver and renal functions were normal. The samples of SAT and VAT were obtained during laparoscopic or open adrenalectomy in patients with PPGL, and during laparoscopic cholecystectomy in controls. VAT samples of patients with PPGL were obtained from the retroperitoneal area. VAT samples of controls were obtained from the omentum. SAT samples were obtained from the anterior abdominal wall.

### 2.2. Anthropometric, Biochemical Measurements and Adipose Tissue Sampling 

Blood samples were collected after overnight fasting between 6 and 7 a.m. Basic laboratory tests were performed by standard methods in our institutional laboratory with international accreditation. A subgroup of patients with diabetes mellitus (DM) was defined by previously diagnosed DM or fasting plasma glucose levels ≥ 7.0 mmol/L or plasma glucose ≥ 11.1 mmol/L two hours after a 75 g oral glucose load or HbA1C ≥ 48 mmol/mol. Obesity was defined by BMI ≥30 kg/m^2^ and overweight by BMI ≥ 25 kg/m^2^. Office arterial blood pressure was measured with an oscillometric sphygmomanometer. Arterial hypertension was defined as systolic blood pressure ≥ 140 mm Hg and/or the diastolic blood pressure ≥ 90 mm Hg, repeatedly.

Plasma free metanephrines (normetanephrine and metanephrine) were quantified by liquid chromatography with electrochemical detection (HLPC-ED, Agilent 1100, Agilent Technologies, Inc., Wilmington, DE, USA). Our norms for metanephrine were 0.140 and 0.540 nmol/L, and norms for normetanephrine were 0.130 and 0.790 nmol/L. Plasma catecholamines were extracted by commercial kit (Catecholamines in plasma O.N.5000., Chromsystems Instruments & Chemicals GmbH, Munich, Germany). Analytes were analyzed by liquid chromatography with electrochemical detection (HPLC-ECD). The tissue samples of VAT and SAT were collected through an RNA stabilization reagent (RNAlater, Qiagen, Hilde, Germany) and stored at −80 °C before further examination. 

### 2.3. Indirect Calorimetry 

Energy metabolism was quantified by indirect calorimetry with a ventilated canopy system (Vmax Encore 29 N system, VIASYS Healthcare Inc; SensorMedics, Yorba Linda, CA, USA). Resting energy expenditure (REE) and respiratory quotient (RQ) were measured. The methodology was described in our previous article [5]. We used the Harris–Benedict formula to calculate the predicted basal energy expenditure (BEE). To express rate of metabolism (REE/BEE), REE was divided by BEE and multiplied by 100. Hypermetabolic state was classified as REE/BEE ≥ 110 %. 

### 2.4. Determination of mRNA Expression by Quantitative Real-Time PCR (RT-PCR)

Total RNA was isolated from SAT and VAT samples. The sample (70 mg) was homogenized on a MagNA Lyser Instrument (Roche Diagnostics GmbH, Mannheim, Germany). Total RNA was automatically isolated from the homogenate on a MagNA Pure instrument using a Magna Pure Compact RNA Isolation kit (tissue) (Roche Diagnostics GmbH, Mannheim, Germany). RNA purity and quantity were determined on a NanoPhotometer (Implen, Munchen, Germany). Reverse transcription of 250 ug of total RNA was performed, using random primers according to the manufacturer’s protocol of the High-Capacity cDNA Reverse Transcription Kits (Applied Biosystems, Foster City, CA, USA). Gene expression determination was performed on a 7500 Real-Time PCR System (Applied Biosystems, Foster City, CA, USA). For reaction, a mix of TaqMan^®^ Universal PCR Master Mix II, NO AmpErase^®^ UNG (Applied Biosystems, Foster City, CA, USA), nuclease-free water (Fermentas Life Science, Vilnius, Lithuania) and specific TaqMan^®^ Gene Expression Assays (Applied Biosystems, Foster City, CA, USA) were used. The catalog numbers for the specific TaqMan^®^ Gene Expression Assays were as follows: *ADRB3* (Hs00609046_m1), *CEBPB* (Hs00270923_s1), *CIDEA* (Hs00154455_m1), *DIO 2* (Hs00255341_m1), *FBXO 31* (Hs00375554_m1), *LHX8* (Hs00418293_m1), *PPARD* (Hs00602622_m1), *PPARG* (Hs00234592_m1), *PPARGC1A* (Hs00173304_m1), *PRDM16* (Hs00223161_m1), *SHOX2* (Hs00243203_m1), *TMEM26* (Hs00415619_m1), *TNFRSF9* (Hs00155512_m1), *UCP1* (Hs01084773_m1), *ZIC1* (Hs00602749_m1) and *B2M* (Hs99999907_m1).

Beta-2-microglobulin (*B2M*) was used as a valid housekeeping gene in this population. LDL receptor related protein 10 (*LRP10*) and T-box 1 (*TBX1*) were also tested as a suitable endogenous gene. The formula 2^-ddCt^ was used to calculate relative gene expression. 

F-box only protein 31 (*FBXO31*), LIM homeobox 8 (*LHX8*) and Zic family member 1 (*ZIC1*) were chosen as markers of BAT. Cluster of differentiation-137 (*CD137*), also known as TNF receptor superfamily member 9 (*TNFSRF9*), transmembrane protein 26 (*TMEM26*) and short stature homeobox 2 (*SHOX2*) were chosen as markers of BeAT [21,22,23,24,25]. Cell death-inducing DFFA-like effector (*CIDEA*) and Iodothyronine deiodinase 2 (*DIO2*) were chosen as markers for both [26]. PR/SET domain 16 (*PRDM16*), CCAAT enhancer binding protein beta (*CEBPB*), peroxisome proliferator activated receptor delta (*PPRD*), peroxisome proliferator-activated receptor gamma (*PPARG*) and PPARG coactivator 1-Alpha (*PPARGC1A*) were chosen as key transcription factors in adipogenesis. Uncoupling protein 1 (*UCP1*) was chosen as the functional marker of non-shivering thermogenesis. 

### 2.5. Statistical Analysis

Statistical analysis was performed by Statistica for Windows ver. 12.1 (StatSoft, Inc., Tulsa, OK, USA). Normally distributed data are shown as the mean ± SD (standard deviation). Data with abnormal distribution are shown as median with interquartile range (IQR). Categorical variables are expressed as frequencies (%). All parameters were tested for normality by the Shapiro–Wilk test. Parameters without a normal distribution were logarithmically converted before use of the parametric test. Two independent groups were tested by the Student’s t-test or Mann–Whitney test, as appropriate. The dependent groups were tested by the Student’s paired *t*-test or the Wilcoxon test, as appropriate. Correlations between variables were investigated by the Pearson correlation coefficient. Categorical variables were tested by chi-square or Fisher’s exact test. *p* values of < 0.05 were considered significant. 

## 3. Results

Clinical and metabolic characteristics of the study subjects are summarized in Table 1. Both groups did not differ significantly in age, BMI, blood pressure measurements or ratio of females and males. Both groups contained subjects of normal weight, overweight and obesity. Nine patients with PPGL suffered from diabetes, three of them were treated with metformin and one dose of insulin, four patients were treated only with metformin. Two patients follow dietary restrictions. No patient was treated by PPARγ-agonists. No patient underwent intensified conventional insulin treatment. All subjects were euthyroid. Two control subjects were treated with a small dose of statins and one control subject was treated with beta blockers because of palpitations. Statins were also part of medications of ten patients with PPGL. No control subject suffered from diabetes. As expected, the PPGL group had higher FBG, HbA1C and free plasma metanephrine levels. They also expressed a higher metabolic rate and levels of HDLc in comparison with the controls. Fifteen patients with PPGL (60%) fulfilled the criteria for hypermetabolism. 

### 3.1. Gene Profile of VAT and SAT of Patients with PPGL and Controls

The mRNA expression of all genes in SAT and VAT of PPGL and controls is summarized in Table 2. In VAT of PPGL, we found markedly increased mRNA expression of *CEBPB* (*p* < 0.001), *DIO2* (*p* = 0.005), *PPARGC1A* (*p* < 0.001) and *UCP1* (*p* < 0.001). The difference between *PRDM16* expression was at the level of significance (*p* = 0.069). On the contrary, we found decreased levels of *TMEM26* in VAT of PPGL (*p* = 0.007). The rest of the genes showed no significant change. In SAT of PPGL, we found increased mRNA expression of adrenoceptor beta 3 (*ADRB3*) (*p* = 0.046), *CIDEA* (*p* = 0.048) and *PPARGC1A* (*p* = 0.001). On the contrary, expression of *FBXO31* was higher in SAT of controls (*p* = 0.035). mRNA expression of *ZIC1* and *LHX8* was undetermined in SAT and VAT of both groups. mRNA expression of *UCP1* was undetermined in SAT of both groups.

### 3.2. Analysis of Gene Expression in Patients with PPGL

Selected correlation analysis between mRNA expression of genes in VAT and SAT of patients with PPGL and their metabolic and clinical characteristics is shown in Table 3 and Figure 1. Complete correlation analysis between mRNA expression of *UCP1* and other genes in VAT of patients with PPGL and their metabolic characteristics is shown in Table 4. As expected, expression of genes related to BAT and BeAT in visceral adipose tissue of patients with PPGL decreased with age and BMI. Contrarily, the expression of those genes positively correlated with plasma norepinephrine and normetanephrine levels. Surprisingly, only expression of one gene correlated with BEE/REE. In SAT of patients with PPGL the situation was similar, but not so apparent. The most obvious was the correlation with BMI. Expression of *UCP1* in VAT of patients with PPGL correlated significantly with the expression of other genes—*ADRB3*, *CEBPB*, *CIDEA*, *DIO 2*, *FBXO 31*, *PPARD*, *PPARGC1A*, *PPARG*, *PRDM16* and *TMEM26*. Expression of *UCP1* in VAT of patients with PPGL also positively correlated with HDLc levels and negatively with TAG levels. Furthermore, HDLc levels showed positive correlation with *ADRB3*, *CIDEA*, *PPARGC1A* and *PRDM16* of VAT. TAG showed negative correlation also with expression of *ADRB3* and *PPARGC1A* in VAT of PPGL. Similar relations were observed in SAT of PPGL. 

## 4. Discussion

We demonstrate in retroperitoneal VAT of patients with PPGL higher expression levels of *UCP1* with positive correlation to plasma norepinephrine levels and its metabolite normetanephrine and mRNA expression of *ADBR3* gene encoding beta3-adrenoceptors. Increasing *UCP1* expression levels also correlated with increasing expression of other genes involved in adipogenesis of BAT/BeAT (*CEBPB*, *CIDEA*, *DIO 2*, *FBXO 31*, *PPARD*, *PPARGC1A*, *PPARG*, *PRDM16)*. Expression of *UCP1*, *ADBR3*, *CIDEA*, *PPARGC1A* and *PRDM16* correlated positively with HDLc levels, and expression of *ADBR3* and *PPARGC1A* negatively correlated with TG levels. Our group of patients with PPGL tended to have higher levels of circulating HDLc in comparison with the control group. We did not find any correlation between epinephrine and its metabolite metanephrine and *UCP1*. Our results implicate positive effect of norepinephrine-induced BAT on lipid metabolism in humans, and are in accordance with published studies. Bartelt already showed that prolonged cold exposure attenuates hyperlipidemia and improves insulin resistance in mice due to the accelerated plasma clearance of triglycerides by BAT [27]. Okamura noticed significant correlation between fractionated urine norepinephrine level and serum HDLc level in 42 patients with PPGL and the decrease of HDLc after adrenalectomy in this group [4]. Calandra and Spergel described elevated fasting plasma unesterified FA levels in four patients with PHEO [28,29]. Turnbull and Krentz showed that levels of FA normalize after alfa and beta-blockade in patients with PHEO [30,31]. Komada did not detect changes in FA or TAG in 13 patients with PHEO before and after surgical treatment [32]. Chechi showed that in patients undergoing coronary artery bypass grafting, the expression of *UCP1* in epicardial adipose tissue correlates with circulating HDLc and TAG levels [33].

Connection between BAT and adiposity was already described by many studies in rodents and humans [34]. Wang showed in 2011 that activated BAT by catecholamines in patients with PPGL is inversely related to central obesity [35]. Wang also showed in 2015 that detectable BAT in tumor-free humans is associated with lower BMI, less subcutaneous fat areas and visceral fat areas, waist circumferences, lower fasting glucose and triglyceride levels, and increased HDLc concentrations [36]. We noticed in our patients with PPGL that similar tendencies–expression of *UCP1* and other genes decreased with increasing BMI. Although our study group showed more than 60% hypermetabolism, we did not find a direct relationship between gene expression and basal energy metabolism parameters determined by indirect calorimetry. The hypermetabolic state is certainly complex, with a number of other factors.

Age is considered as one of important determinants of BAT in the general population. The location and amount of BAT in rodents and humans change during their lifetime. Mass and function of BAT decreases with aging under both thermoneutral conditions and cold stimulation [37]. BAT activity decreases significantly in populations over 60 [38]. The amount of beige adipocytes also decreases with aging [37]. The distribution of WAT changes during aging too (e.g., changing the ratio between SAT and VAT). We noticed negative correlation between expression of *UCP1* (and other genes) with age in our patients with PPGL. We assume that this finding follows the trend in general population. 

Another determinant of BAT activity in the general population is gender. The female gender is generally connected with a higher amount of BAT. Rodriguez-Cuenca determined the morphological and functional features of BAT in male and female rats. In his study, female rats showed more sensitive signals of beta3-adrenoreceptors to norepinephrine and had greater oxygen consumption, higher *UCP1* content, a higher multilocular arrangement, longer cristae, and higher cristae dense mitochondrion in BAT [39]. In our study, we did not detect the difference in genes expressions relative to gender due to the small number of subjects. 

PPGL as a secondary cause of diabetes mellitus is already established with prevalence varying between 35% and 70%, depending on the study [40,41]. Pathophysiological mechanisms include decreased pancreatic insulin secretion, decreased pancreatic glucagon secretion [42], decreased glycogenesis, increased glycogenolysis and gluconeogenesis in the liver, and insulin resistance in muscles [43]. Contrarily, the adult human BAT was identified by the uptake of the glucose tracer during PET/CT examination. Iwen showed in his study that peripheral glucose uptake and insulin sensitivity significantly improved by 20% during cold exposure [44]. Our group of patients with PPGL showed higher levels of FBG and HbA1C in comparison with the control group. On the other hand, we did not find specific connection between the expression of *UCP1* (or other selected genes) with FBG or HbA1C. It may be explained by the results of studies, which indicate that activated human BAT increases oxidation of lipids more than glucose [45]. 

Further gene expression analysis is complicated because of different regions of fat. We found no significant difference in the expression of core murine genes in the VAT of patients with PPGL in comparison with the VAT of controls. We found low or unchanged levels of beige core genes (*TMEM26*, *TNFSRF9* and *SHOX2*) and brown core genes (*ZIC1*, *FBXO31* and *LHX8*). The list of genes is based mostly on in vitro and animal models, and their function is under debate. Different results were published even for patients with PPGL. Di Franco found in the periadrenal BAT and its derived brown adipose stem cells from eight patients with PPGL both classic brown and beige characteristics. In his study, periadrenal BAT and undifferentiated brown stem cells expressed higher levels of *ZIC1* and *TNFSRF9*, but not *LHX8* [46]. Nagano and colleagues found in 11 subjects with PPGL that retroperitoneal brown cells expressed molecular characteristics of classic brown fat rather than beige [47]. They assessed in the perirenal region of PPGL higher expression of *UCP1*, *CIDEA*, *ELOVL3*, *EBF3*, *FBXO31* and *LHX8*, but not *TNFSRF9*, *TBX1* or *TMEM26*, in comparison with seven subjects with non-functional adrenal tumors [47]. Betz and colleagues examined 57 subjects with adrenal tumors (including PPGL). They found in a BAT-positive group higher expression of *UCP1*, *PPARGC1A*, *DIO2*, *ADRB3*, but not *PRDM16* [48]. Frontini and colleagues found in omental fat of 12 patients with PPGL higher expression of *UCP1*, *ADRB3* and *PRDM16*, but not *PPARGC1A* or *DIO2*. These results showed that the specificity and functions of core brown/beige markers are still unclear, and that further studies must be carried out [46,49]. 

We demonstrate in SAT of PPGL higher expression of *PPARGC1A* and *ADRB3*, together with high *CIDEA* levels. However, we did not detect expression of *UCP1* in SAT of PPGL. The expression of *ADRB3* decreased in SAT of PPGL with increasing age, and the expression of *CIDEA*, *PPARGC1A*, *PPARG* and *PRDM16* decreased with increasing BMI. Expression of CIDEA also correlated positively with norepinephrine levels and negatively with TG levels. Whether browning could occur in SAT is questionable and, to date, negative results were published in PPGL [46,50]. Absence of *UCP1* expression did not exclude thermogenesis because the *UCP1*-independent mechanism in BeAT was also described [51]. 

Our study had several limitations. Firstly, both groups included subjects who were overweight or obese, which are states connected with appreciable changes in adipose tissue. Secondly, the VAT samples were not taken from the same regions. We are aware of the difficulties resulting from the distinct origin and localization of various fat depots in the human body. We also examined the whole adipose tissue instead of the derived adipocytes themselves. Our population was also small and of a cross-sectional nature. Thirdly, we cannot exclude the influence of medications. Our patients with PPGL were treated with alpha (doxazosin exclusively) and beta blockers, metformin, and statins. Finally, gene expression is not equal to protein expression.

## 5. Conclusions

In conclusion, we demonstrate signs of *UCP1*-dependent norepinephrine-induced thermogenesis, connected with overexpression of *DIO2*, key transcriptional factors *PPARGC1α*, *CEBPB* and *PRDM16* in retroperitoneal VAT of PPGL, and its relations to circulating HDLc and triglycerides levels. BMI and age were assessed in relation with UCP1 in patients with PPGL, as is stated in the general population. Surprisingly, we do not find a direct relationship between increased basal energy metabolism and gene expression. Furthermore, predominance of BAT or BeAT gene signature in VAT of PPGL was not detected. In SAT of PPGL, we found signs of possible BeAT transformation, but without simultaneously undergoing *UCP1*-dependent thermogenesis.

## Figures and Tables

**Figure 1 biomedicines-10-00586-f001:**
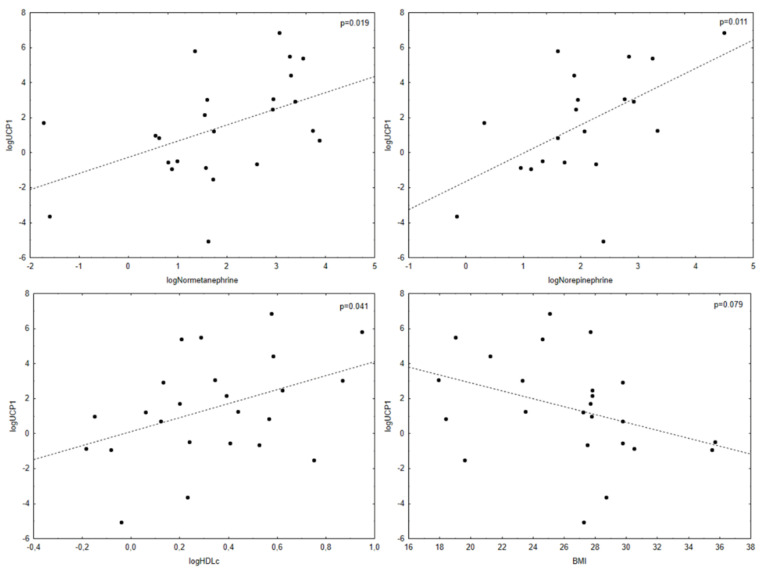
Correlation analysis between *UCP1* expression and plasma normetanephrine, plasma norepinephrine, HDLc levels and BMI.

**Table 1 biomedicines-10-00586-t001:** Clinical and metabolic characteristics of study subjects.

Factor	PPGL	Controls	
	*n* = 25	*n* = 14	*p* Value
Female (*n*, %)	13 (52)	9 (64)	0.872
Age (years)	54.3 ± 14	55.7 ± 15	0.775
Body mass index (kg·m^2^)	26.3 ± 4.7	27.0 ± 3.3	0.628
P-metanephrine (nmol/L)	3.1 [7.4]	0.2 [0.1]	0.001
P-normetanephrine (nmol/L)	5.5 [18.6]	0.3 [0.2]	<0.001
P-norepinephrine (nmol/L)	6.9 [12.1]	-	
P-epinephrine (nmol/L)	1.2 [2.7]	-	
FBG (mmol/L)	5.8 [1.7]	4.9 [0.3]	0.005
HbA1C (mmol/mol)	44.0 [18]	35.0 [4]	<0.001
Total cholesterol (mmol/L)	4.8 ± 1.1	5.2 ± 0.9	0.363
HDLc (mmol/L)	1.5 ± 0.5	1.4 ± 0.5	0.055
LDLc (mmol/L)	2.8 ± 0.9	2.9 ± 0.7	0.983
Triglycerides (mmol/L)	1.3 ± 1.0	2.1 ± 1.6	0.618
Systolic BP (mmHg)	132 ± 18	125 ± 16	0.273
Diastolic BP (mmHg)	82 ± 12	79 ± 13	0.392
REE (Kcal/day)	1690 ± 313	1462 ± 313	0.087
BEE (Kcal/day)	1508 ± 234	1511 ± 192	0.974
REE/BEE (%)	111.8 ± 14	96.1 ± 11	0.008
Diabetes mellitus (*n*, %)	9 (36)	0 (0)	-
Obesity (*n*, %)	3 (12)	3 (21)	-
Alpha blockers (*n*, %)	25 (100)	0 (0)	-
Beta blockers (*n*, %)	18 (72)	1 (8)	-
Statins (*n*, %)	10 (40)	2 (15)	-
Peroral antidiabetics (*n*, %)	7 (28)	0 (0)	-
Insulin (*n*, %)	3 (12)	0 (0)	-

Abbreviations: P-, plasma; FBG, fasting blood glucose; HbA1c; glycated hemoglobin; HDLc, high-density lipoprotein cholesterol; LDLc, low-density lipoprotein cholesterol; BP, blood pressure; REE, resting energy expenditure; BEE, basal energy expenditure; PAD, peroral antidiabetics.

**Table 2 biomedicines-10-00586-t002:** mRNA expression of genes potentially connected with BAT and BeAT in SAT and VAT samples of patients with PPGL and controls.

Gene Symbol	Gene Name	SAT			VAT		
		PPGL	Controls		PPGL	Controls	
		*n* = 25	*n* = 14	*p* Value	*n* = 25	*n* = 14	*p* Value
*ADRB3*	Adrenoceptor Beta 3	1.5 [2.3]	0.4 [0.4]	0.046	2.4 [4.4]	1.2 [1.9]	0.156
*CEBPB*	CCAAT Enhancer Binding Protein Beta	0.9 [0.7]	0.9 [0.6]	0.255	1.3 [0.8]	0.7 [0.5]	<0.001
*CIDEA*	Cell death-inducing DFFA-like effector	1.4 [1.3]	0.8 [0.7]	0.048	1.2 [1.4]	0.9 [0.9]	0.532
*DIO 2*	Iodothyronine Deiodinase 2	1.1 [0.8]	0.76 [1.3]	0.554	1.09 [2.3]	0.47 [0.4]	0.005
*FBXO 31*	F-Box Protein 31	0.91 [0.4]	1.2 [0.3]	0.035	1.0 [0.3]	0.89 [0.3]	0.378
*LHX8*	LIM Homeobox 8	Undet.	Undet.	-	Undet.	Undet.	-
*PPARD*	Peroxisome Proliferator Activated Receptor Delta	0.7 [0.6]	0.9 [0.4]	0.971	1.0 [0.5]	1.0 [0.6]	0.320
*PPARG*	Peroxisome Proliferator-Activated Receptor Gamma	1.0 [0.4]	0.9 [0.5]	0.144	1.3 [0.6]	0.9 [0.7]	0.248
*PPARGC1A*	PPARG Coactivator 1α	1.5 [1.0]	0.6 [0.7]	0.001	1.7 [2.0]	0.5 [0.3]	<0.001
*PRDM16*	PR/SET domain 16	1.1 [0.6]	0.9 [0.4]	0.224	1.0 [0.8]	0.7 [0.5]	0.069
*SHOX2*	Short Stature Homeobox 2	0.9 [1.0]	0.9 [0.6]	0.624	1.1 [1.2]	0.8 [0.8]	0.609
*TMEM26*	Transmembrane Protein 26	1.1 [1.5]	1.1 [1.5]	0.892	0.8 [0.6]	1.1 [1.7]	0.007
*TNFRSF9*	TNF Receptor Superfamily Member 9	0.8 [1.4]	1.5 [1.2]	0.126	0.8 [0.8]	1.2 [0.9]	0.075
*UCP1*	Uncoupling Protein 1	Undet.	Undet.	-	3.5 [25.0]	0.1 [0.7]	<0.001
*ZIC1*	Zic Family Member 1	Undet.	Undet.	-	Undet.	Undet.	-

Abbreviations: Undet., Undetermined.

**Table 3 biomedicines-10-00586-t003:** Correlation analysis between mRNA expression of genes in VAT and SAT, and metabolic parameters of patients with PPGL with *p* value < 0.1.

	VAT				SAT		
Gene Symbol	Factor	R	*p* Value	Gene Symbol	Factor	R	*p* Value
*ADRB3*	Age	−0.398	0.049	*ADRB3*	Age	−0.507	0.016
*CIDEA*	Age	−0.354	0.082	*CIDEA*	BMI	−0.624	<0.001
*PPARG*	Age	−0.418	0.037	*PPARGC1A*	BMI	−0.596	0.002
*UCP1*	Age	−0.347	0.096	*PPARG*	BMI	−0.477	0.016
*ADRB3*	BMI	−0.427	0.033	*PRDM16*	BMI	−0.568	0.003
*CIDEA*	BMI	−0.349	0.087	*CIDEA*	P-Nore	0.488	0.024
*PPARGC1A*	BMI	−0.457	0.021	*CIDEA*	P-Epi	0.375	0.093
*UCP1*	BMI	−0.365	0.079	*PRDM16*	P-Epi	0.452	0.051
*ADRB3*	HDLc	0.512	0.008	*TMEM26*	P-Epi	−0.411	0.060
*CIDEA*	HDLc	0.436	0.029	*CIDEA*	TAG	−0.565	0.003
*PPARGC1A*	HDLc	0.487	0.013	*PPARGC1A*	TAG	−0.534	0.005
*PPARG*	HDLc	0.337	0.098	*PPARG*	TAG	−0.535	0.005
*PRDM16*	HDLc	0.414	0.039	*CIDEA*	HDLc	0.395	0.054
*UCP1*	HDLc	0.418	0.041	*PPARGC1A*	HDLc	0.396	0.049
*ADRB3*	TAG	−0.481	0.014	*ADRB3*	HbA1C	−0.474	0.029
*PPARGC1A*	TAG	−0.560	0.003				
*UCP1*	TAG	−0.380	0.066				
*UCP1*	P-Norm	0.474	0.019				
*DIO 2*	P-Norm	0.530	0.006				
*TMEM26*	P-Nora	0.531	0.013				
*PPARGC1A*	P-Nore	0.480	0.027				
*UCP1*	P-Nore	0.553	0.011				
*DIO 2*	P-Nore	0.541	0.011				
*SHOX2*	BEE/REE	−0.561	0.005				

Abbreviations: BMI, body mass index; P-Epi, plasma epinephrine; P-Norm, plasma normetanephrine; P-Nore, plasma norepinephrine; FBG, fasting blood glucose; REE, resting energy expenditure; BEE, basal energy expenditure; HbA1c; glycated hemoglobin; HDLc, high-density lipoprotein cholesterol; TAG, triglycerides.

**Table 4 biomedicines-10-00586-t004:** Complete correlation analysis between mRNA expression of UCP1 and other genes in VAT of patients with PPGL.

Gene Symbol	R	*p* Value
*ADRB3*	0.711	<0.001
*CEBPB*	0.466	0.003
*CIDEA*	0.741	<0.001
*DIO 2*	0.782	<0.001
*FBXO 31*	−0.410	0.047
*PPARD*	0.329	0.044
*PPARGC1A*	0.832	<0.001
*PPARG*	0.667	<0.001
*PRDM16*	0.688	<0.001
*SHOX2*	0.011	0.949
*TMEM26*	0.434	0.033
*TNFRSF9*	−0.176	0.290

Abbreviations: BMI, body mass index; FBG, fasting blood glucose; HbA1c; glycated hemoglobin; HDLc, high-density lipoprotein cholesterol; LDLc, low-density lipoprotein cholesterol; REE, resting energy expenditure.

## Data Availability

The data presented in this study are available on request from the corresponding author. The data are not publicly available due to privacy and ethical principles.

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
