# Peer review of "Gene Profile of Adipose Tissue of Patients with Pheochromocytoma/Paraganglioma"

_biomedicines, 2022, doi:10.3390/biomedicines10030586_

Round 1
Reviewer 1 Report
Klimova et al. present a study on genes expression in adipose tissue of patients with PPGL.
Major points:
- Controls have to be more extensively described in the materials and methods section
- Adipose tissue from PPGL patients and controls weren’t obtained from the same region, which may induce a major bias in the study
- To little genes were explored in the different groups to conclude: only three for BAT, only five for BeAT… and in the Results section these genes expressions are discordant. The authors have to explore more genes.
- 3 patients take insulin which would interfered with the results, these patients should be exclude of the study, and the detail of peroral antidiabetics should be given.
- In Wu et al. CEBP is up regulated in peri adrenal adipose tissue and PPARG is down regulated. How do the authors explain the difference with there study?
Minors points:
- In the abstract section, the controls should be described
- Many abbreviations are not defined in the main text
- CD173 is not in the Taqman assays genes list
- Patients consent’s should be written in the materials and methods section
- Norms of metanephrine and normetanephrine should be given in the main text
- Did FDG PET was performed in the PPGL patients and in the controls?
- Genetic status of the PPGL patients should be given.
Author Response
Dear Reviewer,
Thank you for your comments, suggestions, and the opportunity to improve our manuscript. The main text was modified. Specific answers follow.
Controls have to be more extensively described in the materials and methods section.
Information about control subjects were added into main text (lines 118 – 121 and 206 - 208). Control group included patients undergoing elective cholecystectomy in stable phase of disease without concomitant inflammation. Control subjects did not differ in gender or age. Control subjects did not suffer from any severe chronic disease, oncology or otherwise. Their liver and renal functions were normal. Three control subjects were obese (three patients with PPGL were also obese). Two control subjects were treated by small dose of statins and one patient was treated with small dose of beta blocker because of palpitations. Statins were also part of medications of ten patients with PPGL. No control subject suffered from diabetes.
Adipose tissue from PPGL patients and controls weren’t obtained from the same region, which may induce a major bias in the study.
Different regions of the visceral fat samples are one of the limitations of our study. The first region of visceral fat relates to little activity of brown or beige adipose tissue in health or disease based on available imaging and laboratory studies, even in patients with pheochromocytoma. The second depot of visceral fat is rich in activity and/or incidence of brown/beige fat. For the reasons mentioned above, we did not compare the occurrence of the brown/beige adipose tissue in those two depots. We tried to identify the genetic background of brown/beige adipose tissue surrounding functional pheochromocytomas, avoiding genes related to white visceral adipose tissue. We compared our findings with other studies using similar methodology in patients with those catecholamine-producing tumours.
To little genes were explored in the different groups to conclude: only three for BAT, only five for BeAT… and in the Results section these genes expressions are discordant. The authors have to explore more genes.
Genetic background of brown and beige adipose tissue in human and animals is still matter of debate. The list of genes related to brown or beige adipose tissue is based mostly on in vitro and animal models and their function is also under debate. Research groups often chose different sets of genes, and even if they chose the same gene panel their results are not always in harmony. Those studies show us, how difficult the topic is. In the beginning, we determined more genes in our study, but our knowledge about their function is even more limited, so our interpretation could be misleading. For example, some other results follow:
|
VAT |
SAT |
|||||
|
PPGL |
Controls |
p value |
p value |
|||
|
DDIT3 |
-0.045 ± 0.5 |
0.114 ± 0.6 |
0.257 |
0.014 ± 0.3 |
-0.002 ± 0.3 |
0.891 |
|
NRIP1 |
0.014 ± 0.7 |
-0.151 ± 0.5 |
0.433 |
0.089 ± 0.4 |
-0.230 ± 0.7 |
0.093 |
|
ERN1 |
-0.070 ± 0.5 |
0.119 ± 0.5 |
0.267 |
0.082 ± 0.5 |
-0.168 ± 0.2 |
0.022 |
|
RXRG |
6.60 ± 1.1 |
6.615 ± 0.1 |
0.985 |
0.985 ± 0.5 |
0.721 ± 0.5 |
0.124 |
3 patients take insulin which would interfered with the results, these patients should be exclude of the study, and the detail of peroral antidiabetics should be given.
The part of the manuscript was modified (lines 206 – 209). Pheochromocytoma and functional paraganglioma are one of the secondary causes of diabetes mellitus with prevalence varying between 35 and 70 % depending on study. In our study, 9 patients (36 %) with PPGL suffered from diabetes, three of them were treated with metformin and one dose of insulin, four patients were treated only with metformin. Two patients follow dietary restrictions. No patient was treated by PPARγ-agonists. No patient underwent intensified conventional insulin treatment. Half a year after the operation (tumorectomy) the glucose metabolism in majority of patients has normalized. For this reason, we did not excluded patients treated by insulin before the operation. We are aware that the heterogenous treatment of patients with PPGL is one of the limitations of the study.
As you suggested, we recalculated majority of genes after exclusion of 3 patients who are being treated with insulin, results follow. Only ADRB3 in SAT showed significant change and TMEM26 in VAT subtle change (red colour).
|
Gene symbol |
|
SAT |
|
VAT |
|||
|
|
PPGL |
Controls |
PPGL |
Controls |
|||
|
|
n = 22 |
n = 14 |
p value |
n = 22 |
n = 14 |
p value |
|
|
ADRB3 |
0.4 [1.5] |
-0.9 [1.1] |
0.032 |
0.8 [1.7] |
0.2 [2.2] |
0.208 |
|
|
CEBPB |
0.05 [0.7] |
-0.1 [0.7] |
0.224 |
0.2 [0.7] |
-0.3 [0.9] |
<0.001 |
|
|
CIDEA |
0.4 [1.3] |
-0.2 [0.8] |
0.039 |
0.2 [1.3] |
-0.2 [1.0] |
0.605 |
|
|
LHX8 |
Undet. |
Undet. |
- |
Undet. |
Undet. |
- |
|
|
PPARD |
0.008 [0.6] |
-0.1 [0.5] |
0.790 |
0.05 [0.4] |
-0.01 [0.7] |
0.358 |
|
|
PPARG |
-0.008 [0.4] |
-0.2 [0.5] |
0.182 |
0.2 [0.6] |
-0.1 [0.6] |
0.340 |
|
|
PPARGC1A |
0.5 [0.7] |
-0.5 [1.0] |
0.001 |
0.5 [1.4] |
-0.7 [0.7] |
<0.001 |
|
|
PRDM16 |
0.2 [0.7] |
-0.01 [0.4] |
0.260 |
-0.02 [0.7] |
-0.3 [0.7] |
0.082 |
|
|
SHOX2 |
-0.1 [1.0] |
-0.1 [0.5] |
0.715 |
0.02 [0.9] |
-0.3 [0.8] |
0.665 |
|
|
TMEM26 |
0.2 [1.2] |
0.06 [1.1] |
0.832 |
-0.2 [0.5] |
0.08 [1.1] |
0.016 |
|
|
TNFRSF9 |
-0.4 [1.7] |
0.3 [0.8] |
0.117 |
-0.1 [01.0] |
1.2 [0.9] |
0.172 |
|
|
UCP1 |
Undet. |
Undet. |
- |
1.2 [3.6] |
0.2 [0.6] |
<0.001 |
|
|
ZIC1 |
Undet. |
Undet. |
- |
Undet. |
Undet. |
- |
|
(Results are here logarithmic)
In Wu et al. CEBP is up regulated in peri adrenal adipose tissue and PPARG is down regulated. How do the authors explain the difference with there study?
Wu et al. compared in “Role of PDK4 in insulin signaling pathway in periadrenal adipose tissue of pheochromocytoma patients” peri adrenal adipose tissue of patients with pheochromocytoma and peri adrenal adipose tissue of patients with non-functional adenomas. As you said, they found higher mRNA levels of PGC1α or C/EBPβ and lower mRNA levels of PPARγ in patients with pheochromocytoma. We had similar result in C/EBPβ and PGC1α (both are upregulated in visceral fat of our patients with PPGL). Difference in expression of PPARγ was not significant in our study. We consider PPARγ as a less specific for beige or brown adipogenesis.
In the abstract section, the controls should be described
The section was modified (lines 50 – 51).
Many abbreviations are not defined in the main text
More abbreviations were explained in the main text.
CD173 is not in the Taqman assays genes list
We are sorry for this misunderstanding. We used different name for CD137: TNF receptor superfamily member 9 (TNFSRF9). This name was listed in TaqMan® Gene Expression Assays (lines 172 and 180).
Patients consent’s should be written in the materials and methods section.
The Ethical Committee of our institution approved the study (permission date: 21 May 2015, ethical code: 20/15). The study was done in accordance with the Declaration of Helsinki (line 387 – 389).
All patients signed Patients consent’s, which was requested with the manuscript itself.
Norms of metanephrine and normetanephrine should be given in the main text
The norms were added. Our norms for metanephrine were 0.140 and 0.540 nmol/l and norms for normetanephrine were 0.130 and 0.790 nmol/l (line 140 – 141).
Did FDG PET was performed in the PPGL patients and in the controls?
No, FDG PET/CT was performed only in PPGL patients.
Genetic status of the PPGL patients should be given.
All patients underwent genetic testing for germline mutations. Patients with germline mutations were excluded from the study.
Yours faithfully,
O.Petrák and J. Klímová, on behalf of all authors

Reviewer 2 Report
Klimova et al. have investigated mRNA expression profiles for brown and beige adipose tissue in subcutaneous and visceral fat of patients with a catecholamine-producing pheochromocytoma or paraganglioma and in controls without such a tumor. The study is well presented and limitations have been stated clearly in the discussion.
A few minor points that should be clarified and improved:
- You used three different housekeeping genes, which is good practice. Could you please add, whether the results were similar when using the other 2 housekeeping genes, and if not, why you decides for B2M.
- Could you clarify whether blood sampling was performed after 20 minutes supine rest for all patients, or whether there were differences in the sampling procedure between the group with pheochromocytoma and controls?
- I find the presentation of correlation in Table 3 and 4 a little confusing. In Table 3, the legend says, “selected genes”, wouldn’t it be better to show all genes with significant correlations? If this already the case, please rephrase the legend accordingly.
Furthermore, there are duplications in Table 3 and 4, please remove redundancy. Are the three columns on the right of Table 4 necessary? All significant correlations are already part of Table 3 (two of them with slightly different p-values than table 3), and the non-significant ones don’t need to go in the main text. You could have a supplemental table with all calculations.
Author Response
Dear Reviewer,
Thank you for your comments, suggestions, and the opportunity to improve our manuscript. Specific answers follow.
You used three different housekeeping genes, which is good practice. Could you please add, whether the results were similar when using the other 2 housekeeping genes, and if not, why you decides for B2M.
Housekeeping genes were chosen by prof. Haluzík's laboratory based on their best experience. One part of our study was analysing also immune actions and cells in our samples of fat. The results are not given here, but B2M was chosen as housekeeping gene because of that.
Could you clarify whether blood sampling was performed after 20 minutes supine rest for all patients, or whether there were differences in the sampling procedure between the group with pheochromocytoma and controls?
Only metanephrines were taken in supine position. All other laboratory tests were performed in upright position. Laboratory tests were usually performed in two sessions, so the sampling procedure between the group with pheochromocytoma and controls was, at least, similar.
I find the presentation of correlation in Table 3 and 4 a little confusing. In Table 3, the legend says, “selected genes”, wouldn’t it be better to show all genes with significant correlations? If this already the case, please rephrase the legend accordingly.
Furthermore, there are duplications in Table 3 and 4, please remove redundancy. Are the three columns on the right of Table 4 necessary? All significant correlations are already part of Table 3 (two of them with slightly different p-values than table 3), and the non-significant ones don’t need to go in the main text. You could have a supplemental table with all calculations.
We rephrased two legends, hopefully is clearer now (lines 253 – 254 and lines 259 – 260). The non-significant values in Table 4 were included on request of one of superior colleague, who asked for complete results including UCP1 and its full correlation. We reduced them now.
Yours faithfully,
O.Petrák and J. Klímová, on behalf of all authors
Round 2
Reviewer 1 Report
I would like to thank the authors for their answers.
I have one more concern :
In the answer the authors write : "All patients underwent genetic testing for germline mutations. Patients with germline mutations were excluded from the study."
and in the main text : "Only one patient had positive testing for germline mutation (TMEM127 c.159del (p.Trp53CysfsTer28)."
The authors must clarify this part.
Author Response
Dear Reviewer,
We apologize for the conflicting information about genetic in the text and thank you for your comment.
In fact, no patient suffered from von Hippel-Lindau syndrome, multiple endocrine neoplasia type 2 or neurofibromatosis type 1. No patient had mutation in SDHx or MAX gene. One patient had positive testing for germline mutation in TMEM127 c.159del (lines 117 – 120 in the manuscript).
Yours sincerely,
O.Petrák and J. Klímová, on behalf of all authors